# COMPARATIVE KNOWLEDGE DISTILLATION

## ABSTRACT

In the era of large-scale pretrained models, Knowledge Distillation (KD) serves an important role in transferring the wisdom of computationally-heavy teacher models to lightweight, efficient student models while preserving performance. Traditional KD paradigms, however, assume readily available access to teacher models for frequent inference—a notion increasingly at odds with the realities of costly, often proprietary, large-scale models. Addressing this gap, our paper considers how to minimize the dependency on teacher model inferences in KD in a setting we term Few-Teacher-Inference Knowledge Distillation (FTI-KD). We observe that prevalent KD techniques and state-of-the-art data augmentation strategies fall short in this constrained setting. Drawing inspiration from educational principles that emphasize learning through comparison, we propose Comparative Knowledge Distillation (CKD), which encourages student models to understand the nuanced differences in a teacher model's interpretations of samples. Critically, CKD provides additional learning signals to the student without making additional teacher calls. We also extend the principle of CKD to groups of samples, enabling even more efficient learning from limited teacher calls. Empirical evaluation across varied experimental settings indicates that CKD consistently outperforms state-of-the-art data augmentation and KD techniques.

## 1 INTRODUCTION

The growing demand for smaller models that retain the capabilities of large pretrained ones has spurred interest in efficient compression techniques. Though Knowledge Distillation (Hinton et al., 2015) stands out as a promising solution approach, the escalating parameter count in teacher models significantly drives up inference costs, whether in API charges or computational resource time. This naturally raises the question: can we perform KD with minimal teacher calls?

KD is often performed either by learning to imitate the teacher's representation of a single sample (often requiring many such representations to learn effectively) (Hinton et al., 2015) or by augmenting the samples to ask *additional* questions of the teacher (Beyer et al., 2022) — paradigms that are inefficient with respect to the number of teacher calls. Additional learning paradigms have been applied to KD (Tian et al., 2019; Zheng et al., 2022), yet none are designed for enhancing learning outcomes *with limited teacher calls*, a setting we refer to as "Few-Teacher-Inference Knowledge Distillation" (FTI-KD).

To solve this problem, we take inspiration from the field of education, in which a foundational learning method is *learning by comparison* (Rittle-Johnson & Star, 2011). This style of pedagogy attempts to capture not just the teacher's solution to a single problem, but the nuanced comparison between different problems (Rittle-Johnson & Star, 2009).

This paper introduces Comparative Knowledge Distillation (CKD): a novel learning paradigm that seeks to bring this intuition to Knowledge Distillation by encouraging the student's difference in representation between samples to mimic the teacher's difference in representation between the same samples. Unlike data augmentation techniques used for KD such as Mixup (Zhang et al., 2017; Beyer et al., 2022), CKD enables teacher representations to be computed on samples and then combined later, minimizing the number of queries to the teacher.

We investigate CKD's performance in KD experiments with limited teacher calls. Across different image classification architectures, number of teacher calls, and the depth of access to the teacher model (intermediate outputs vs. logits-only), CKD consistently improves upon state-of-the-art data

augmentation and knowledge distillation techniques, improving performance over the next highest method by over **4%** absolute top-1 accuracy over the next highest method and, for some resource levels, by up to **7%**. Our code is publicly available.[1]

## 2 RELATED WORK

There are four closely related areas in Knowledge Distillation to our work: KD-Specific loss functions, Data Augmentation Strategies for KD, Relational KD approaches, and Contrastive Learning.

**KD-Specific Loss Functions**    Starting with Hinton et al. (2015)'s KL divergence loss between teacher and student losses, many papers built different loss functions specific to KD (Huang & Wang, 2017; Peng et al., 2019; Ahn et al., 2019; Passalis & Tefas, 2018). Many works have also applied KD to intermediate layer representations when given "white box" access to the teacher model's intermediate representations (Haidar et al., 2021; Wu et al., 2021; Shu et al., 2021; Li et al., 2023; Zhang et al., 2022). Comparative Knowledge Distillation is *complementary* to these approaches, as these loss functions can be applied to our comparative representations as well as to sample representations.

**Data Augmentation**    Data augmentations such as flipping, cropping, rotating, and cutout have set the state of the art on some KD tasks (Xu et al., 2020; Yang et al., 2021; Fu et al., 2020; DeVries & Taylor, 2017) and aggregating these strategies together has shown promise as well (Cubuk et al., 2018). Augmentation strategies based on Mixup (Zhang et al., 2017) have been particularly performant (Wang et al., 2022; Liang et al., 2020) and synthetic data generation techniques have enabled KD in extremely low-resource settings (Wang, 2021; Nguyen et al., 2022; Wang et al., 2020). Data augmentation strategies can be very effective at augmenting the amount of data that can be used to query the teacher, but in the FTI-KD setting teacher calls are limited due to the cost of teacher queries. By contrast, CKD is designed to add additional learning signal *without additional teacher calls*.

**Relation-Based KD**    In relation-based KD losses, a student's learning signal is derived from a distance metric applied to both the student and the teacher's representations of a pair or group of samples. Many methods implement variants of this approach (Park et al., 2019; Liu et al., 2019; Peng et al., 2019; Dai et al., 2021), some applying these methods across or within representation channels (Gou et al., 2022; Huang et al., 2022) or within prediction classes (Huang et al., 2022). Relation-Based KD losses are similar to CKD in that they compare student and teacher representations of groups of samples, but different in that they collapse the representation space into a single number: often euclidean distance or angle between vectors (Park et al., 2019). To the best of our knowledge, no existing KD approaches have considered learning from high dimensional comparisons between groups of samples.

**Contrastive Learning**    Contrastive Learning approaches for KD such as CRD (Tian et al., 2019) and ReKD (Zheng et al., 2022) represent a different but related approach to cross-sample learning from ours. Contrastive Learning methods encourage the student's representation of one sample to be similar or different to the teacher's representation of another, depending on whether the two samples are considered a "positive" or "negative" pair by a pseudolabelling function that may require ground truth labels (Tian et al., 2019). This is similar to our method in that representations from multiple samples are involved, but different in the objective we optimize. CKD encourages students to match a teacher's *comparison* between two samples by having the student consider both samples itself, and requires no pseudolabelling (i.e., positive and negative pairs).

## 3 COMPARATIVE KNOWLEDGE DISTILLATION

The core problem addressed in this paper is Few-Teacher-Inference Knowledge distillation (FTI-KD). In this FTI-KD setting, only few teacher calls are possible, constraining the amount of data the student can use for KD training. The intuition of Comparative Knowledge Distillation (CKD)

---

[1]bit.ly/ComparativeKD

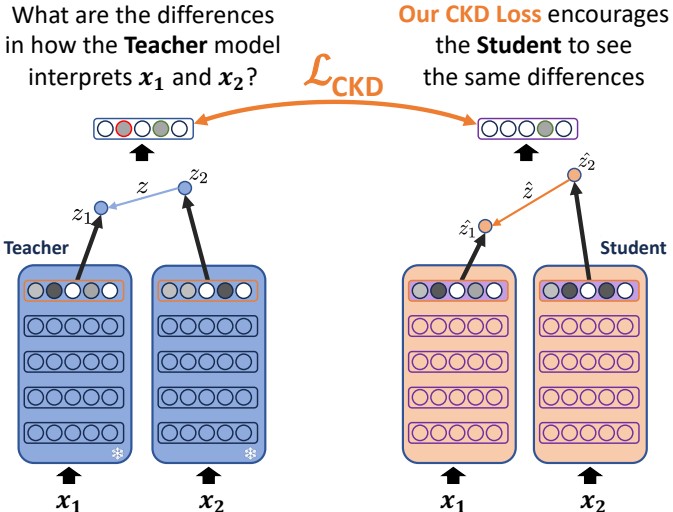

Figure 1: Comparative Knowledge Distillation (CKD): a novel training paradigm that encourages student and teacher representations of the *differences between sample representations* to be similar. Critically, because teacher representations can be cached and recombined into many possible comparisons, CKD offers an additional learning signal *without requiring additional calls to the teacher*.

is that instead of distilling knowledge by encouraging a student to mimic a teacher's output on a single sample, we would like to encourage the student to mimic the teacher's *comparison of two or more different samples*. We hypothesize that capturing the nuances of how the teacher interprets the *similarities and differences between* samples may prove may provide a strong training signal for the student in this low-resource setting. Our method is illustrated in Figure 1.

### 3.1 NOTATION AND PROBLEM FORMULATION

The FTI-KD setting assumes that we can make at most $n$ calls to a "teacher" model, a large, performant model on this task, receiving teacher representations $z_i$ in return for samples $x_i$. In KD, these $z$ values are usually logit representations, although in the the "white-box" case (Romero et al., 2014), they are intermediate layer representations. KD settings commonly attempt to encourage the student's representation $\hat{z}_i$ to be similar to $z_i$. As in other KD settings (Hinton et al., 2015; Tian et al., 2019) we assume access to ground truth labels for these samples $y_i$.

### 3.2 CKD LOSS FUNCTION FOR $k = 2$ SAMPLES

CKD is a loss function that encourages the comparison of the student's representation of two or more samples to be similar to the teacher's comparison of those samples. We implement comparison as the vector difference operation in order to effectively capture nuanced comparison information between representations. In order to optimize the Kullback–Leibler divergence loss as is common in KD (Hinton et al., 2015), we pass both the student and teacher differences through the softmax function to output probability distributions.

$$\hat{p}_\Delta = \text{softmax}(\hat{z}_i - \hat{z}_j) \tag{1}$$
$$p_\Delta = \text{softmax}(z_i - z_j) \tag{2}$$
$$\mathcal{L}_{CKD} = \mathcal{L}_{KL}(\hat{p}_\Delta || p_\Delta) \tag{3}$$

The final loss function is a combination of cross-entropy loss between student logit representations and the ground truth outputs and our proposed CKD loss. These losses are linearly combined to form a differentiable loss, weighted by hyperparameter $\beta$.

$$\mathcal{L} = \mathcal{L}_{CE} + \beta \mathcal{L}_{CKD} \tag{4}$$

Table 1: CKD consistently outperforms state-of-the-art KD and data augmentation techniques across various low-resource settings and teacher-student combinations.

| $n$ | 1600 | 2000 | 2400 | 2800 | 3200 |
|---|---|---|---|---|---|
| **WRN-40-2→WRN-16-2** | | | | | |
| KD (Hinton et al., 2015) | $26.09_{2.75}$ | $32.71_{1.96}$ | $34.97_{2.52}$ | $39.34_{3.44}$ | $43.05_{1.92}$ |
| RKD (Park et al., 2019) | $22.92_{3.61}$ | $28.07_{2.05}$ | $32.11_{2.00}$ | $37.34_{2.49}$ | $39.69_{0.82}$ |
| Dist (Huang et al., 2022) | $26.73_{1.97}$ | $30.62_{0.80}$ | $35.51_{3.12}$ | $38.86_{0.65}$ | $42.92_{0.59}$ |
| Mixup (Zhang et al., 2017) | $27.20_{0.69}$ | $31.30_{0.49}$ | $34.10_{0.41}$ | $37.33_{0.58}$ | $39.33_{0.88}$ |
| CRD (Tian et al., 2019) | $29.37_{2.17}$ | $35.40_{1.61}$ | $38.41_{0.45}$ | $42.06_{2.47}$ | $45.34_{1.32}$ |
| CKD | $\mathbf{36.38_{0.60}}$ | $\mathbf{39.21_{1.38}}$ | $\mathbf{43.27_{0.40}}$ | $\mathbf{47.81_{1.11}}$ | $\mathbf{50.14_{1.36}}$ |
| **VGG13→VGG8** | | | | | |
| KD (Hinton et al., 2015) | $28.85_{0.80}$ | $33.34_{0.59}$ | $35.97_{0.32}$ | $38.67_{0.84}$ | $41.39_{1.25}$ |
| RKD (Park et al., 2019) | $25.63_{0.99}$ | $28.51_{0.80}$ | $31.93_{1.48}$ | $36.20_{2.16}$ | $37.79_{0.86}$ |
| Dist (Huang et al., 2022) | $29.09_{0.55}$ | $32.31_{1.65}$ | $35.89_{2.88}$ | $38.38_{0.75}$ | $41.54_{2.74}$ |
| Mixup (Zhang et al., 2017) | $25.93_{0.35}$ | $29.32_{0.32}$ | $31.77_{0.64}$ | $33.70_{0.60}$ | $36.19_{0.16}$ |
| CRD (Tian et al., 2019) | $30.14_{0.97}$ | $33.87_{0.87}$ | $36.59_{0.38}$ | $40.26_{0.53}$ | $42.48_{0.48}$ |
| CKD | $\mathbf{33.04_{0.41}}$ | $\mathbf{36.95_{0.53}}$ | $\mathbf{40.14_{0.62}}$ | $\mathbf{43.07_{0.30}}$ | $\mathbf{44.34_{0.23}}$ |
| **Resnet110→Resnet32** | | | | | |
| KD (Hinton et al., 2015) | $24.87_{0.31}$ | $30.14_{2.20}$ | $32.84_{1.74}$ | $39.68_{4.57}$ | $39.15_{1.25}$ |
| RKD (Park et al., 2019) | $19.05_{0.56}$ | $24.04_{2.03}$ | $30.97_{5.79}$ | $33.20_{1.35}$ | $39.84_{0.20}$ |
| Dist (Huang et al., 2022) | $23.17_{0.61}$ | $28.22_{2.36}$ | $31.50_{1.44}$ | $35.05_{1.37}$ | $42.71_{2.20}$ |
| Mixup (Zhang et al., 2017) | $24.41_{1.49}$ | $27.29_{2.19}$ | $31.99_{1.52}$ | $32.98_{1.48}$ | $35.89_{1.04}$ |
| CRD (Tian et al., 2019) | $26.06_{2.00}$ | $33.91_{1.56}$ | $36.63_{1.35}$ | $40.50_{0.99}$ | $44.38_{1.45}$ |
| CKD | $\mathbf{32.47_{2.63}}$ | $\mathbf{38.46_{0.78}}$ | $\mathbf{41.98_{1.58}}$ | $\mathbf{46.16_{0.94}}$ | $\mathbf{45.90_{1.57}}$ |

## 3.3 EXTENSION TO $k \geq 2$ SAMPLES

One important property of CKD is that it enables students to learn from these comparisons *without additional teacher calls*, unlike augmentation techniques such as Mixup which benefit from calling the teacher repeatedly on different augmentations of the input (Beyer et al., 2022). In the $k = 2$ formulation above, CKD can add comparisons for all combinations of two samples in the dataset of $n$ teacher calls. This is $\binom{n}{2}$ comparisons, which is $O(n^2)$. If we were able to learn from the teacher's "difference" between three, four, or ...$k$ samples, the student would have exponentially more ($O(n^k)$) comparisons to learn from.

Motivated by this intuition, we extend CKD to settings with $k > 2$ in the following way: we randomly split the $k > 2$ samples into two groups, aggregate the representations of the samples within each group, and compare the group representations.

We introduce the following additional notation: we term the teacher and student representations of the samples in each group as $Z_A, \hat{Z}_A$ and $Z_B, \hat{Z}_B$, and we define an aggregation function $\gamma : \mathbb{R}^{a \times D} \to \mathbb{R}^D$ which maps a group of $a$ representations to a single representation for that group. We choose a simple $\gamma$ in our implementation, the centroid function.

CKD loss is then determined as above, this time with the aggregated representations of each group.

$$\hat{P}_\Delta = \text{softmax}(\gamma(\hat{Z}_A) - \gamma(\hat{Z}_B)) \tag{5}$$

$$P_\Delta = \text{softmax}(\gamma(Z_A) - \gamma(Z_B)) \tag{6}$$

$$\mathcal{L}_{CKD} = \mathcal{L}_{KL}(\hat{P}_\Delta || P_\Delta) \tag{7}$$

Intuitively, we expect that there may be an optimal setting of $k$ for each experimental setting. As $k$ increases, so too will the amount comparative samples to learn from. Yet, because the centroid function can be seen as an interpolation that regularizes the logit manifold (Zhang et al., 2020), higher values of $k$ will also have group representations that may be overly smoothed, losing important information useful for training.

## 4 EXPERIMENTAL SETUP

### 4.1 METHODOLOGY

We construct the Few-Teacher-Inference KD setting by constraining the KD experimental setting from Tian et al. (2019) to allow for limited teacher calls $n$.

**Limited Teacher Calls**   We conduct our experiments on the commonly used CIFAR-100 dataset. We investigate limited teacher call settings by constraining the dataset to randomly chosen subsets ($n$) in the range $[1600, 4800]$ by increments of 400. We split the data 80-20% for train and validation and evaluate on the CIFAR-100 test set.

**Teacher-Student Combinations**   We also explore various teacher-student combinations motivated by prior KD works (Tian et al., 2019), WRN-40-2 to WRN-16-2, Resnet110 to Resnet32, and VGG13 to VGG8.

**Data Preprocessing**   When passing samples through any model (teacher or student), we perform a random cropping of 32x32 with a padding of 4, followed by a random horizontal flip as in Tian et al. (2019). We randomly select $n$ samples for the few-teacher-inference (FTI-KD) setting. As in previous work, we assume access to ground truth labels for the samples we query from the teacher.

**Training Details**   We run each student model over three trials and report the mean and standard deviation of our results. We train to convergence using early stopping on the validation loss. We use early stopping instead of fixed epochs so that each algorithm runs to convergence before evaluation. We use trained teacher models from Tian et al. (2019). Each run takes between 20 ($n = 1600$) and 40 minutes ($n = 4800$) on a single 12 GB consumer GPU – we primarily use a 2080Ti for our experiments. We describe additional training details in Appendix A.

### 4.2 BASELINES

We report results on the following baselines, selected because of their strong performance on KD tasks and their data augmentation properties in low-resource settings.

1. **Knowledge Distillation (KD)** (Hinton et al., 2015): this is the standard KD loss, employing KL divergence loss between the teacher and student logits.

2. **Contrastive Representation Distillation (CRD)** (Tian et al., 2019) is a contrastive learning method that uses the label to group "positives" and "negatives" in each batch and encourage the student's representations to be similar to the teacher's for positives and dissimilar for negatives.

3. **Mixup**: As Mixup applied to KD requires additional teacher calls on the mixed up inputs (Liang et al., 2020), we implement the "Fixed Teacher" (Beyer et al., 2022) version of data augmentation, in which the teacher's output logits from the original datapoints are recombined and used for supervision.

4. **Relational Knowledge Distillation (RKD)** (Park et al., 2019) is a "Relational KD" approach based on learning a distance metric over the teacher's relationship between two samples. By contrast, our proposed CKD encourages students to match *high dimensional* relations from the teacher by attempting to match the vector difference between samples. Additionally, CKD scales to larger groups of samples, $k = 3, 4...$, by aggregating intra-group representations.

5. **Distillation from a Stronger Teacher (DIST)** (Huang et al., 2022) is a recently proposed relational approach that works particularly well in cases where the teacher model is much stronger than the student. DIST improves over the standard KD loss by considering the cross-sample relations and encourages the student to match the intra-class probabilities with the teacher across samples.

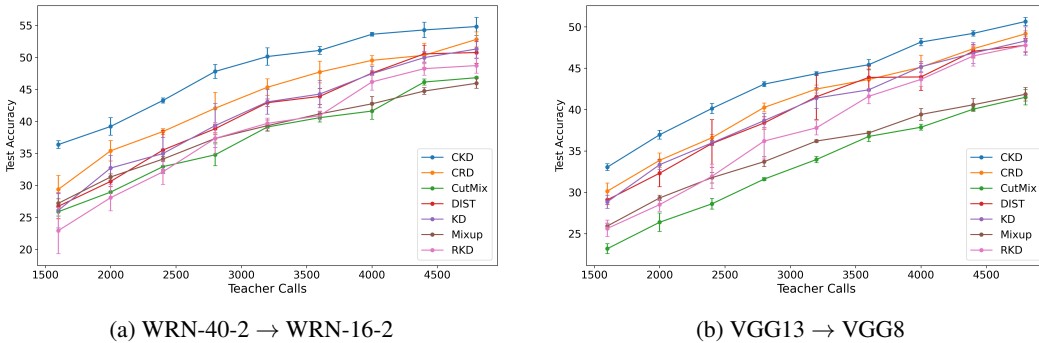

(a) WRN-40-2 → WRN-16-2    (b) VGG13 → VGG8

Figure 2: Results from Table 1 represented visually for WRN and VGG models. CKD consistently outperforms baselines across low-resource teacher calls on different teacher-student distillation settings common in the literature (Tian et al., 2019) Points and error bars are the mean and standard deviation of runs over three random seeds.

### 4.3    EXTENSION TO WHITE-BOX SETTING

One common KD setting is "white-box", in which not only are the teacher-produced logits available for training, but so too are the teacher model's intermediate layer outputs for those samples. Some KD loss functions are designed specifically for intermediate layer distillation. Our approach is *complementary* to these; we simply replace the teacher and student representations of a single sample with the teacher and student's representations of the difference between two samples. In our experiments, we demonstrate this by combining CKD with two widely used intermediate layer losses, FitNets (Romero et al., 2014) and Variational Information Distillation (VID) (Ahn et al., 2019) and investigating whether CKD brings performance improvements.

## 5    RESULTS AND DISCUSSION

### 5.1    KD RESULTS

Our results are depicted visually in Figure 2 and numerically in Table 1. We find that across a variety of student-teacher combinations, including wide resnet (WRN), VGG, and Resnet models, our approach consistently outperforms baselines on the FTI-KD setting.

Using the wide resnet models (WRN) as teacher and student, CKD outperforms the next highest performing method, CRD (Tian et al., 2019) consistently. Comparing the mean across trials and across all low-resource $n$ ranging from 1600 to 4800, CKD outperforms CRD 47.85% to 43.44% on top-1 accuracy, an improvement of **4.41%** absolute accuracy. This difference is even more pronounced in lower resource settings; when $n \in \{1600, 2000, 2400\}$ CKD outperforms CRD by **7.01 3.81, 4.86**, and **5.75%**. On average across all $n$, CKD outperforms other methods by wide margins as well, including KD (**6.83%**), RKD (**9.61%**), DIST (**7.01%**), and Mixup (**9.64%**).

Results are similarly encouraging for the VGG and Resnet110 distillation settings, although slightly less pronounced. Averaged across all $n$ for VGG models, CKD outperforms KD, RKD, and Mixup baselines by **3.34%**, **5.71%**, **8.34%**, outperforms the next-best method, CRD, by **2.49%**, and the next closest method DiST by **3.46%**. And for Resnet110 models averaged across all $n$, CKD outperforms the next best approach CRD by **3.15%**, KD by **6.39%**, DIST by **7.54%**, RKD by **8.98%**, and Mixup by **10.19%**.

Although our results show strong improvements over the baselines, this constrained FTI-KD setting is difficult for all methods. Teacher models perform above 70%, leaving plenty of room for future research to adddress this problem.[2] Numerical results for larger values of $n$ are in Appendix B.

---

[2]The trained Resnet110, WRN-40-2, and VGG13 teacher models achieve 74.32% 75.59%, and 74.64% top-1 test accuracy respectively.

Table 2: Given white-box access to intermediate teacher outputs, CKD seamlessly integrates with KD losses designed to learn from intermediate representations, improving their performances.

| Method | 1600 | 2400 | 3200 | 4000 | 4800 |
|---|---|---|---|---|---|
| **WRN-40-2→WRN-16-2** | | | | | |
| FitNets (Romero et al., 2014) | $24.02_{0.90}$ | $30.73_{5.10}$ | $39.70_{1.94}$ | $45.45_{2.13}$ | $48.04_{1.12}$ |
| +CKD | $\mathbf{36.20_{1.02}}$ | $\mathbf{43.16_{2.81}}$ | $\mathbf{48.79_{0.63}}$ | $\mathbf{52.41_{0.77}}$ | $\mathbf{54.79_{1.12}}$ |
| VID (Ahn et al., 2019) | $28.72_{1.80}$ | $36.73_{1.19}$ | $42.49_{1.58}$ | $48.34_{1.31}$ | $51.32_{0.99}$ |
| +CKD | $\mathbf{35.85_{0.29}}$ | $\mathbf{43.37_{1.20}}$ | $\mathbf{50.43_{0.63}}$ | $\mathbf{53.38_{1.03}}$ | $\mathbf{55.77_{0.89}}$ |
| **VGG-13→VGG-8** | | | | | |
| FitNets (Romero et al., 2014) | $26.46_{1.40}$ | $34.20_{1.90}$ | $39.78_{1.04}$ | $43.52_{1.79}$ | $47.35_{0.66}$ |
| +CKD | $\mathbf{29.81_{1.35}}$ | $\mathbf{36.44_{0.64}}$ | $\mathbf{41.64_{0.95}}$ | $\mathbf{44.81_{1.04}}$ | $\mathbf{48.54_{0.85}}$ |
| VID (Ahn et al., 2019) | $29.05_{1.74}$ | $35.83_{1.37}$ | $40.46_{1.24}$ | $45.07_{1.31}$ | $48.69_{0.82}$ |
| +CKD | $\mathbf{31.41_{0.87}}$ | $\mathbf{40.18_{0.44}}$ | $\mathbf{45.31_{0.55}}$ | $\mathbf{48.21_{0.98}}$ | $\mathbf{50.35_{0.40}}$ |

## 5.2 EXTENSION TO WHITE-BOX ACCESS

We find that CKD also integrates with different intermediate layer loss functions seamlessly, improving two commonly used intermediate layer loss functions by substantial margins. Our results are depicted in Table 2. In the WRN distillation setting, adding CKD to Fitnets leads to an improvement of **12.43%** absolute top-1 accuracy improvement. On average across low resource teacher calls $n$ ranging from 1600 to 4800, CKD led to a **9.45%** absolute accuracy improvement. Results of adding CKD to VID were similar although not quite as pronounced, leading to a **6.24%** absolute accuracy improvement. On the VGG models, the margins were tighter although no less consistent, leading an average improvement of **1.99%** and **3.27%** for FitNets and VID respectively. We believe these results indicate that CKD can be complementary with intermediate layer losses.

Table 3: We find that the choice of comparison function is meaningful: comparing samples based on the vector difference between their representations consistently outperforms addition and interpolation.

| $n$ | 1600 | 2000 | 2400 | 2800 | 3200 |
|---|---|---|---|---|---|
| + | $32.93_{0.32}$ | $37.68_{0.91}$ | $42.16_{0.62}$ | $44.7_{0.73}$ | $47.25_{1.87}$ |
| $\lambda$ | $31.91_{1.83}$ | $37.88_{2.87}$ | $41.38_{2.46}$ | $45.36_{0.91}$ | $46.97_{1.71}$ |
| - | $\mathbf{36.38_{0.60}}$ | $\mathbf{39.21_{1.38}}$ | $\mathbf{43.27_{0.40}}$ | $\mathbf{47.81_{1.11}}$ | $\mathbf{50.14_{1.36}}$ |

## 5.3 ABLATIONS ON COMPARISON FUNCTION AND SAMPLES

To analyze why our method outperforms the baselines, we investigate the role of the two critical hyperparameters of our method: the comparison function and the number of points to be compared, $k$.

**Comparison Functions** The goal of the comparison function is to determine a nuanced metric of how a teacher model compares two sample representations (or sample-group representations if $k > 2$). The simplest and most intuitive of these is the vector difference operation, which literally addresses the question: *how does the teacher interpret these samples differently?* However, we also consider two other comparison functions: interpolation and addition. In general, the comparison functions we consider can be generalized as

$$\phi(a, b) = \lambda_1 a + \lambda_2 b \tag{8}$$

where our difference comparison can be seen as setting $(\lambda_1, \lambda_2) = (1, -1)$, addition as $(1, 1)$, and interpolation as $(\alpha, 1 - \alpha)$, where $\alpha$ is drawn at random from the $\beta(1, 1)$ distribution, as in (Zhang et al., 2017).

We experiment with these different comparison functions on the WRN models, setting $k$ to the best performing value $k = 3$, and report our results in Table 3. The difference function outperforms alternatives, bringing improvements of up to 2-3% absolute accuracy. We hypothesize that this may be due to the intuition presented in Figure 1 – by encouraging students to understand how the teacher views the differences between two sample representations, we encourage students to learn meaningful nuances of the teacher's representation space that may not be captured in single-sample loss optimization.

**The Role of Number of Comparison Samples $k$** We also investigate how the choice of $k$ impacts performance for four different low-resource settings of $n$, across WRN and VGG models. Intuitively, as we explain in Section 3.3, we expect that there will be an optimal setting of the hyperparameter $k$; as $k$ increases, it will add more comparative samples data for training, but those samples will be increasingly regularized because of the centroid interpolation between larger clusters of $k/2$ datapoints. We find there is generally a "hump" in performance as expected, centered around $k = 3$. This is visualized in Figure 3.

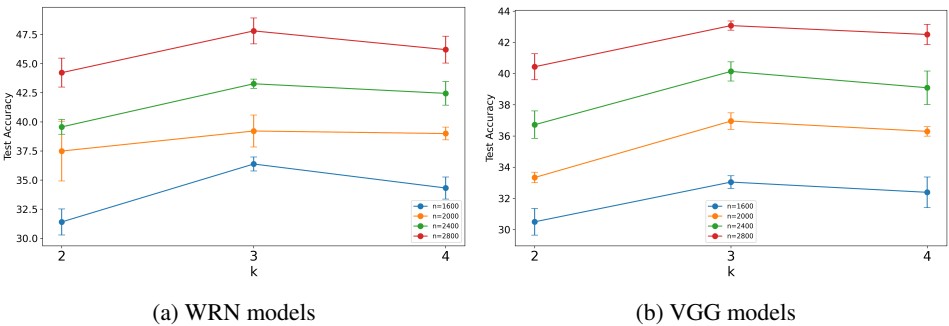

(a) WRN models        (b) VGG models

Figure 3: In line with the intuition presented in Section 3, we find that there is an optimal setting of $k$. As $k$ increases, the amount of comparisons increase, but they are also increasingly regularized by the aggregation function $\gamma$.

## 5.4 ANALYZING THE REPRESENTATIONS LEARNED BY CKD

There are two core challenges in the FTI-KD setting: matching the teacher's representation (the "KD" challenge) and learning from low-resource examples, which is often seen as a generalization challenge. We reproduce two experiments from related works to explore how CKD handles these challenges.

Table 4: Training with CKD leads to an improvement in matching the student's correlation across class logits to the teacher's, a property CRD (Tian et al., 2019) found important for KD representation learning. This table depicts the average absolute difference of student and teacher's correlation matrices; lower is better. Surprisingly, CKD outperforms even CRD, which *explicitly* optimizes this objective.

| Teacher
Student | Resnet110
Resnet32 | VGG13
VGG8 | WRN-40-2
WRN-16-2 |
|---|---|---|---|
| Mixup (Zhang et al., 2017) | 0.162 | 0.148 | 0.154 |
| RKD (Park et al., 2019) | 0.102 | 0.097 | 0.094 |
| DIST (Huang et al., 2022) | 0.107 | 0.093 | 0.095 |
| KD (Hinton et al., 2015) | 0.094 | 0.088 | 0.092 |
| CRD (Tian et al., 2019) | 0.097 | 0.094 | 0.094 |
| CKD | **0.084** | **0.087** | **0.082** |

**Student-Teacher Logit Correlations** Tian et al. (2019) showed that capturing the inter-class correlations between teacher logits is important to successful KD outcomes in students. We reproduce

the experiment from (Tian et al., 2019) to analyze how well CKD encourages this desirable property in students: the details are described below.

Across 100 randomly chosen samples from the CIFAR-100 test set, we first calculate the correlation matrices between class logits for both the teacher and the student. This is done by centering the data by mean, computing the outer product of the resulting vectors to arrive at the covariance matrix, then normalizing by standard deviation to yield the correlation matrix. Then, we report the average absolute difference between the student (trained in different ways) and the teacher's correlation matrices. Lower is better, because a value of 0 would indicate perfect imitation of the teacher's inter-class logit correlations.

In Table 4 we report the numerical results of this correlation analysis from (Tian et al., 2019). Our method outperforms baselines including CRD (Tian et al., 2019), whose objective *explicitly* attempts to capture inter-class correlations. This analysis, along with the main results, indicates that CKD's comparative loss function is providing strong KD learning outcomes.

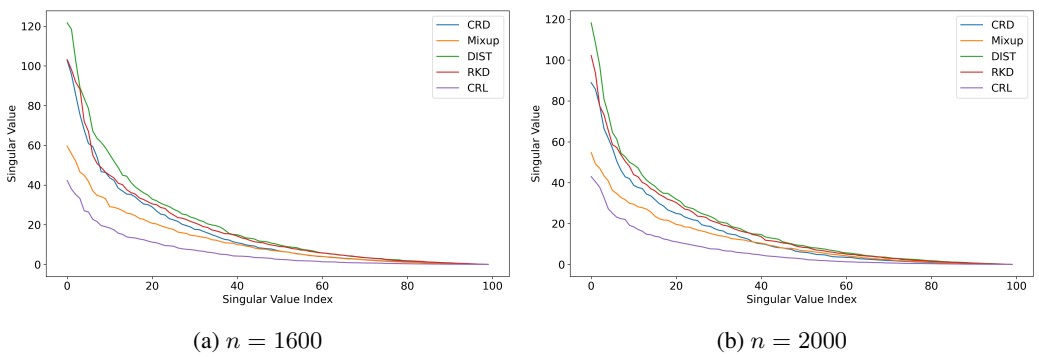

(a) $n = 1600$                 (b) $n = 2000$

Figure 4: CKD acts as a regularizer, flattening models' representation spaces: a property that is closely tied to generalization (Tishby & Zaslavsky, 2015; Shwartz-Ziv & Tishby, 2017).

**CKD Flattens the Representation Space**    A second intuition is that CKD may act as a regularizer, introducing an additional learning signal that helps shape the optimization space in ways that are favorable to generalizable learning of the teacher model under low-resource conditions. To investigate this, we analyze the *flatness* of class representations space, which has been linked to generalization by established theory (Tishby & Zaslavsky, 2015; Shwartz-Ziv & Tishby, 2017). We do this by performing the analysis from Verma et al. (2019) which analyzes the flatness of the representations by performing Singular Value Decomposition (SVD) on the representations, where a lower curve indicates flatter representations. We perform this experiment across two low-resource settings of $n$ on the saved WRN student models' logit representations. Our results are visualized in Figure 4 – CKD's SVD curve is substantially below others, indicating that CKD may act as a regularizer, promoting generalization in the challenging low-resource FTI-KD setting.

## 6    CONCLUSION

In this paper we introduced Comparative Knowledge Distillation (CKD), a novel learning paradigm that we show is useful in performing Knowledge Distillation from few-teacher calls (FTI-KD). CKD does this by augmenting existing teacher calls into comparative samples and defining a loss that encourages student models to mimic teacher's difference in representation between samples. Empirical evaluations reveal CKD's superiority over state-of-the-art KD techniques across various settings. Moreover, with access to intermediate teacher outputs, CKD is complementary to specially designed KD loss functions. CKD achieves these results in part because it captures critical inter-class correlations and acts as a regularizer on the logit space, enhancing generalization in the low-resource setting. This study sets a foundation for future KD research in the era of large-scale pretrained models. One important limitation of this line of research is a deeper understanding of when and how biases in teacher models can be inherited by student models. Future work may find a principled investigation of bias transfer in knowledge distillation fruitful and foundational for understanding the broader implications of KD research.

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

# A  TRAINING DETAILS

## A.1  GENERAL TRAINING PROCEDURE

The batch size was set to 64 in training and the temperature parameter in the KD loss was set to 4 as in Tian et al. (2019). During training, for each epoch, we ensured that each method was trained the same number of steps. The number of steps in an epoch is equal to the number of original samples (images and labels from the CIFAR-100) dataset. All experiments were run on three random seeds: $\{1, 2, 3\}$. Three learning rates were searched for each setting: $\{0.1, 0.05, 0.025\}$, centered around the default of $0.05$ in Tian et al. (2019). The best learning rate was chosen for each setting of number of teacher calls and model by picking the learning rate that yielded the highest mean top-1 accuracy on the validation set across the three trials.

We perform learning rate decay three times for each method, with decay rate set to 0.1, conditioned on early stopping convergence with patience set to 50 steps. When continuing training after learning rate decay, we resume from the model with the highest validation accuracy previously. We use the SGD optimizer with a momentum of 0.9 and weight decay of $5 \times 10^{-4}$ for all experiments.

We performed no search over $\beta$, the tradeoff hyperparameter between $\mathcal{L}_{CE}$ and $\mathcal{L}_{KD}$ or $\mathcal{L}_{CKD}$. We set $\beta$ to 1 for simplicity for CKD and keep default values from each of the other works (Tian et al., 2019).

One important note about seeds: each trial uses the *same* random seed for each method so that *both* the model weights and dataset split are initialized the same way.

## A.2  METHOD-SPECIFIC

### A.2.1  CKD DETAILS

Groups are always split evenly and randomly. In the $k = 3$ case, there are two original samples in group A and one sample in group B. The number of data points sampled from the training set was limited to $100,000$ in all experiments.

### A.2.2  MIXUP DETAILS

Mixup was implemented with the $\lambda$ sampled every batch according to a uniform distribution between 0 and 1, as in the default setting of Zhang et al. (2017). Mixup using three samples was also implemented as a baseline, where three weights were sampled independently from a uniform random distribution between 0 and 1, and normalized. This consistently underperformed Mixup, likely because interpolating in the input space between three images would overregularize the input. The number of data points (pairs or triplets) sampled from the training set was also limited to $100,000$ in all experiments.

### A.2.3  RELATIONAL METHODS

All relational methods were implemented with the loss function applied on the output logits. This was to ensure a fair black-box comparison across all our methods, so each have access to the same representation: the logits. The sampling methods for relational methods (if there was a unique sampler) were adapted from the official implementations of the specific technique. The hyperparameters for those unique samplers are also set to their respective default values in the original implementation.

Table 5: Full numerical results on larger values of $n$ (continued in Table 6 below).

| $n$ | 1600 | 2000 | 2400 | 2800 | 3200 |
|---|---|---|---|---|---|
| **WRN-40-2→WRN-16-2** | | | | | |
| KD (Hinton et al., 2015) | $26.09_{2.75}$ | $32.71_{1.96}$ | $34.97_{2.52}$ | $39.34_{3.44}$ | $43.05_{1.92}$ |
| RKD (Park et al., 2019) | $22.92_{3.61}$ | $28.07_{2.05}$ | $32.11_{2.00}$ | $37.34_{2.49}$ | $39.69_{0.82}$ |
| Dist (Huang et al., 2022) | $26.73_{1.97}$ | $30.62_{0.80}$ | $35.51_{3.12}$ | $38.86_{0.65}$ | $42.92_{0.59}$ |
| Mixup (Zhang et al., 2017) | $27.20_{0.69}$ | $31.30_{0.49}$ | $34.10_{0.41}$ | $37.33_{0.58}$ | $39.33_{0.88}$ |
| CutMix (Yun et al., 2019) | $21.74_{1.66}$ | $26.58_{0.61}$ | $31.46_{0.46}$ | $34.11_{0.61}$ | $35.87_{2.79}$ |
| CRD (Tian et al., 2019) | $29.37_{2.17}$ | $35.40_{1.61}$ | $38.41_{0.45}$ | $42.06_{2.47}$ | $45.34_{1.32}$ |
| CKD | $\mathbf{36.38_{0.60}}$ | $\mathbf{39.21_{1.38}}$ | $\mathbf{43.27_{0.40}}$ | $\mathbf{47.81_{1.11}}$ | $\mathbf{50.14_{1.36}}$ |
| **VGG13→VGG8** | | | | | |
| KD (Hinton et al., 2015) | $28.85_{0.80}$ | $33.34_{0.59}$ | $35.97_{0.32}$ | $38.67_{0.84}$ | $41.39_{1.25}$ |
| RKD (Park et al., 2019) | $25.63_{0.99}$ | $28.51_{0.80}$ | $31.93_{1.48}$ | $36.20_{2.16}$ | $37.79_{0.86}$ |
| Dist (Huang et al., 2022) | $29.09_{0.55}$ | $32.31_{1.65}$ | $35.89_{2.88}$ | $38.38_{0.75}$ | $41.54_{2.74}$ |
| Mixup (Zhang et al., 2017) | $25.93_{0.35}$ | $29.32_{0.32}$ | $31.77_{0.64}$ | $33.70_{0.60}$ | $36.19_{0.16}$ |
| CutMix (Yun et al., 2019) | $22.73_{0.61}$ | $25.47_{0.89}$ | $27.56_{0.73}$ | $30.40_{0.34}$ | $32.71_{0.52}$ |
| CRD (Tian et al., 2019) | $30.14_{0.97}$ | $33.87_{0.87}$ | $36.59_{0.38}$ | $40.26_{0.53}$ | $42.48_{0.48}$ |
| CKD | $\mathbf{33.04_{0.41}}$ | $\mathbf{36.95_{0.53}}$ | $\mathbf{40.14_{0.62}}$ | $\mathbf{43.07_{0.30}}$ | $\mathbf{44.34_{0.23}}$ |
| **Resnet110→Resnet32** | | | | | |
| KD (Hinton et al., 2015) | $24.87_{0.31}$ | $30.14_{2.20}$ | $32.84_{1.74}$ | $39.68_{4.57}$ | $39.15_{1.25}$ |
| RKD (Park et al., 2019) | $19.05_{0.56}$ | $24.04_{2.03}$ | $30.97_{5.79}$ | $33.20_{1.35}$ | $39.84_{0.20}$ |
| Dist (Huang et al., 2022) | $23.17_{0.61}$ | $28.22_{2.36}$ | $31.50_{1.44}$ | $35.05_{1.37}$ | $42.71_{2.20}$ |
| Mixup (Zhang et al., 2017) | $24.41_{1.49}$ | $27.29_{2.19}$ | $31.99_{1.52}$ | $32.98_{1.48}$ | $35.89_{1.04}$ |
| CutMix (Yun et al., 2019) | $20.86_{0.84}$ | $26.09_{1.44}$ | $29.97_{0.72}$ | $32.76_{0.29}$ | $36.75_{1.54}$ |
| CRD (Tian et al., 2019) | $26.06_{2.00}$ | $33.91_{1.56}$ | $36.63_{1.35}$ | $40.50_{0.99}$ | $44.38_{1.45}$ |
| CKD | $\mathbf{32.47_{2.63}}$ | $\mathbf{38.46_{0.78}}$ | $\mathbf{41.98_{1.58}}$ | $\mathbf{46.16_{0.94}}$ | $\mathbf{45.90_{1.57}}$ |

### A.2.4 WHITE-BOX METHODS

When continuing from a previous step after adjusting the learning rate, the other trainable modules used in these loss functions are also restored to the state of that previous step. The hyperparameters for these loss functions are set to their default values from Tian et al. (2019).

## B FULL NUMERICAL RESULTS

Table 6: Results on larger values of $n$.

| $n$ | 3600 | 4000 | 4400 | 4800 |
|---|---|---|---|---|
| **WRN-40-2→WRN-16-2** | | | | |
| KD (Hinton et al., 2015) | $44.27_{2.15}$ | $47.46_{1.16}$ | $49.97_{0.95}$ | $51.32_{1.51}$ |
| RKD (Park et al., 2019) | $40.89_{0.65}$ | $46.17_{1.30}$ | $48.24_{1.03}$ | $48.73_{1.20}$ |
| Dist (Huang et al., 2022) | $43.91_{1.25}$ | $47.56_{0.34}$ | $50.56_{1.33}$ | $50.77_{1.70}$ |
| Mixup (Zhang et al., 2017) | $41.17_{0.43}$ | $42.75_{1.16}$ | $44.74_{0.53}$ | $45.96_{0.78}$ |
| CutMix (Yun et al., 2019) | $39.19_{0.93}$ | $41.31_{0.71}$ | $43.63_{1.50}$ | $45.01_{1.33}$ |
| CRD (Tian et al., 2019) | $47.72_{1.70}$ | $49.55_{0.73}$ | $50.33_{1.87}$ | $52.82_{1.18}$ |
| CKD | $\mathbf{51.09_{0.63}}$ | $\mathbf{53.62_{0.30}}$ | $\mathbf{54.30_{1.20}}$ | $\mathbf{54.83_{1.43}}$ |
| **VGG13→VGG8** | | | | |
| KD (Hinton et al., 2015) | $42.38_{0.15}$ | $45.17_{0.63}$ | $46.82_{1.24}$ | $48.30_{1.71}$ |
| RKD (Park et al., 2019) | $41.60_{0.90}$ | $43.65_{0.83}$ | $46.48_{1.23}$ | $47.77_{0.75}$ |
| Dist (Huang et al., 2022) | $43.88_{1.37}$ | $43.94_{1.65}$ | $47.04_{0.82}$ | $47.77_{0.85}$ |
| Mixup (Zhang et al., 2017) | $37.17_{0.11}$ | $39.41_{0.72}$ | $40.58_{0.76}$ | $41.85_{0.82}$ |
| CutMix (Yun et al., 2019) | $34.18_{0.35}$ | $36.64_{0.43}$ | $38.30_{0.77}$ | $39.38_{0.65}$ |
| CRD (Tian et al., 2019) | $43.63_{1.27}$ | $45.13_{1.43}$ | $47.35_{0.23}$ | $49.15_{0.36}$ |
| CKD | $\mathbf{45.42_{0.62}}$ | $\mathbf{48.16_{0.45}}$ | $\mathbf{49.21_{0.33}}$ | $\mathbf{50.64_{0.49}}$ |
| **Resnet110→Resnet32** | | | | |
| KD (Hinton et al., 2015) | $45.75_{1.51}$ | $44.59_{2.22}$ | $47.35_{2.79}$ | $49.74_{1.46}$ |
| RKD (Park et al., 2019) | $41.70_{3.90}$ | $45.90_{1.41}$ | $46.76_{3.21}$ | $49.35_{2.41}$ |
| Dist (Huang et al., 2022) | $44.65_{3.86}$ | $43.91_{3.47}$ | $45.86_{2.69}$ | $48.70_{2.36}$ |
| Mixup (Zhang et al., 2017) | $38.02_{2.07}$ | $40.85_{1.53}$ | $43.09_{0.49}$ | $45.43_{0.38}$ |
| CutMix (Yun et al., 2019) | $39.72_{1.37}$ | $40.19_{1.85}$ | $43.24_{2.12}$ | $43.41_{0.58}$ |
| CRD (Tian et al., 2019) | $48.36_{2.03}$ | $49.28_{2.39}$ | $50.88_{1.15}$ | $53.34_{0.93}$ |
| CKD | $\mathbf{48.63_{1.53}}$ | $\mathbf{51.94_{1.04}}$ | $\mathbf{52.29_{2.36}}$ | $\mathbf{53.82_{0.77}}$ |

