# OpenReview forum: "Comparative Knowledge Distillation"
_ICLR.cc/2024/Conference — ICLR 2024 Conference Withdrawn Submission_

### Official Review · Reviewer_fQMq · 2023-10-31

**Soundness:** 1 poor
**Presentation:** 2 fair
**Contribution:** 2 fair
**Rating:** 3
**Confidence:** 4

**Summary:**

This paper presents Comparative Knowledge Distillation (CKD), a method to perform knowledge distillation through comparisons in the setting of a limited number of teacher inference calls. The core idea is to formulate a k-wise distillation loss that compares the teacher and student representations of k different data points via self-similarity. The claim is that by having $k$-wise distillation loss function, the student can have exponentially more $\mathcal O(n^k)$ comparisons to learn from.

**Strengths:**

* Improving knowledge distillation in the setting of a constrained number of teacher calls is well-motivated.
* The idea to obtain more information from a limited number of points through k-wise comparisons is interesting.

**Weaknesses:**

* The paper lacks clarity in key parts of the presentation. For example, the method itself is not clear: for $k=2$ in the stochastic optimization setting, are the pairwise comparisons done on a per-batch setting?
* The method is not well-motivated or explained in context of prior works in Relational Knowledge Distillation. The loss here is KL divergence between $\mathrm{softmax}(\hat z_i - \hat z_j)$ and $\mathrm{softmax}(z_i - z_j)$, and the subtraction is motivated in Sec. 5.3. But, why does taking the KL divergence of the softmaxed difference of logits even make sense here? Why does, for instance, the squared difference between the normed differences, e.g., $$\left(\frac{||\hat z_i - \hat z_j||}{||\hat z_i|| ||\hat z_j||} - \frac{||z_i - z_j||}{|| z_i ||||z_j||}\right)^2$$ not make sense? How is the proposed approach novel and better than prior metrics used in relational KD?
* My biggest qualm is that the entirety of the work is supported only by evaluations on CIFAR100, and no other datasets. Favorable evaluations on other data sets such as ImageNet and other tasks, such as language, would significantly improve the paper. As it stands, the claims of the paper rest solely on CIFAR100.

**Questions:**

1. How does this method fare on other datasets besides CIFAR100, such as ImageNet? What about beyond vision tasks?
2. What is the justification for the softmax nonlinearity and KL divergence as a comparator metric? How does it compare with other ways to measure differences between the logits?

---

### Official Review · Reviewer_xswU · 2023-11-02

**Soundness:** 3 good
**Presentation:** 4 excellent
**Contribution:** 3 good
**Rating:** 6
**Confidence:** 5

**Summary:**

The authors propose a relation-based KD loss, CKD, that is efficient and performs well with minimal teacher calls (the few-teacher inference KD setting). CKD achieves this by minimizing the difference between the student and teacher's high-dimensional representation.

The authors demonstrate that their method CKD consistently outperforms other KD methods in the minimum teacher call regime using a simple centroid function including in the white-box setting. The authors similarly mention that the difference between representations is more effective than addition or interpolation.

**Strengths:**

The empirical results reported by CKD are impressive in the few-teacher inference setting considering how straightforward the loss formulation is. I found that the manuscript is well-written and does a good job of explaining the CKD method. This paper provides further evidence that KD approaches that mimic teacher outputs are not an effective distillation method, and supports relation-based/contrastive KD efforts for distillation.

I similarly find this idea of few-teacher-inference knowledge distillation an interesting direction of research, and it is a nice contribution to have coined this term in the paper. Overall, the paper was interesting to read and naturally raised interesting clarification and next-step questions.

**Weaknesses:**

My biggest concern is that this work is eerily similar to the relational knowledge distillation work (Park et al., 2019) barring the high-dimensional KD loss in CKD. I can appreciate that having a higher-dimensional loss formulation might be effective in tightly capturing relationships between two samples as compared to a single number as is the case in [1], however, I would have expected there to be further analysis on why this single differentiating factor elicits such a big change in empirical performance. I feel that there should be more emphasis on this aspect, and I find that Section 5.3 with the comparison function hints towards this direction, if you had to employ the same comparison function employed by [1], you should retrieve RKD in its complete form.

To address these concerns, I would encourage the authors to offer some justifications in the paper as to why a higher dimensional comparison between groups of samples is that effective in FTI-KD.

Minor concerns:

- Could the authors provide results that are averaged by more than three trials? Could you do more than 10 trials? Especially in Figure 2.

- In Table 1, I assume the value and the subscript represent the mean and the variance. Could this be made clear in the caption?

- In Figure 3, could you plot the x-axis beyond k>4 to get a better idea of the trend? Could you also average this over more than 3 trials?

[1] Park, W., Kim, D., Lu, Y. and Cho, M., 2019. Relational knowledge distillation. In Proceedings of the IEEE/CVF conference on computer vision and pattern recognition (pp. 3967-3976).

**Questions:**

Clarification questions:

- It's not entirely clear to me why teacher inference would be computationally or economically expensive. Inference in general is cheap, and fast. In comparison to jointly training a teacher and student model, having access to a teacher's inference is already a substantial improvement in computational resource time.  Could you provide more clarity on this?

- How do you select the samples for each group to compute the centroid, are samples randomly selected or there is a method that is employed?

- In Table 2, is there a reason why other methods beyond CKD were omitted in the white-box analysis?

---

### Official Review · Reviewer_P6qh · 2023-11-02

**Soundness:** 3 good
**Presentation:** 3 good
**Contribution:** 2 fair
**Rating:** 3
**Confidence:** 4

**Summary:**

This paper proposes Comparative Knowledge Distillation (CKD) which is a simple and easy-to-understand method for knowledge transfer. CKD simply makes the student learn the logic difference from the teacher. Experiments show promising results under special settings.

**Strengths:**

1. This paper is well-organized and clearly written.
2. Comparison with different methods are reported.

**Weaknesses:**

1. Some statements are over-claimed in this paper. For example, it claims "a novel learning paradigm ...  mimic the teacher’s difference in representation between the same samples." Actually, it is just the relation-based knowledge distillation which distills the relations between samples from the teacher. It looks me that this learning paradigm is not novel.
2. Experiments are conducted in special settings where the baselines are pretty low, e.g., CRD has the performance of 29.37 on CIFAR with the given setting. The experiments are expected to conduct in normal settings for fair and convincing comparison. Currently, the experimental results are not convincing to me.

**Questions:**

1. What is the difference between the proposed framework from the existing relation-based knowledge distillation?
2. Why not compare the proposed method with the baselines in normal settings where they are well-benchmarked?

---

### Official Review · Reviewer_WFsY · 2023-11-03

**Soundness:** 2 fair
**Presentation:** 2 fair
**Contribution:** 2 fair
**Rating:** 3
**Confidence:** 5

**Summary:**

The paper proposes a new method for knowledge distillation: instead of the student trying to mimic the direct output of the teacher, it mimics the difference between two samples (the student's difference should match the teacher's difference). The authors claim that their method is better suited in resource constrained situations where one can only get teacher's outputs a few times. They compare their proposed method (CKD) to different baselines and show improved results on CIFAR100 dataset.

**Strengths:**

1. The method proposed is simple, efficient and is shown to perform better than the baselines on the few-shot scenarios by a good margin (Table 1/2).

2. The authors have done a nice analysis experiment to investigate the learnt representations by the student, showing that the correlations between the student's outputs is very similar to that of the teacher's outputs, even though this was never explicitly part of the objective. (Table 4)

**Weaknesses:**

1. The authors have motivated the Few-Teacher-Inference (FTI) setting and have proposed a method for that setting. However, there is no explanation for why the proposed distillation objective is better suited for it compared to the other baselines. This is especially relevant because there have been works which have tried to distill the *relationship* between the samples; beyond RKD, which the paper has cited and compared, there are also [1, 2, 3] (I am not asking the authors to compare to these). Given that these works are similar in spirit, what is different about the particular objective, beyond its empirical effectiveness?

2. Continuing along the previous point, it is not clear why this method is relevant for few-shot settings specifically. What happens if you use this objective for the typical KD setting (same as Tian et al.) where you do have all the samples from the training set? That is, performing distillation using CKD on all the training examples. Does it still perform better than the baselines?

3. I think there is a slight confusion in what the authors are providing as the motivation for Few-Teacher-Inference vs their actual experimental setup. Limited teacher call means that you only want to get teacher's responses on few-images. But the student can still be trained on the remaining examples with ground-truth (one-hot) labels. However, the experimental setup seems to be dealing with *few-shot* training scenario, in which you only have access to few training images. The authors should clarify this and present results according to their motivation in the beginning. For example, a sample experimental setup will look something like this: typical KD setup  = distilling on 50,000 images. Few-Teacher-Inference setup = distilling on 2000 images, **normal training** (L_CE; Eq. 4) on the remaining 48,000 images.

4. Across the paper, as mentioned above, the explanation for many choices are not clear. For example, in Sec 5.3, the explanation for why difference, as opposed to interpolation (or other variant) of two output vectors performs better is not clear (Page 8; Table 3). Overall, it is difficult for a reader to get a comprehensive understanding for why things are working. As a sample question which is left unexplored - the objective makes the student mimic the difference. However, multiple outputs could have had the same difference; i.e., there is a loss of information. Yet, this loss does not hinder student's performance (and increases it). Why should it be this way?

5. Even beyond this, I do not think the very specific problem is practically useful. It will be useful in case you want to use Few-Teacher-Inference (FTI) to match the performance of teacher inference on all the samples. That is, the knowledge that you inherit from the teacher using CKD is the similar whether you use 1000 teacher inferences or 10,000 teacher inferences.


References

[1] Knowledge Distillation via Instance Relationship Graph. Liu et al. CVPR 2019.

[2] Similarity-Preserving Knowledge Distillation. Tung et al. arXiv 2019.

[3] Correlation Congruence for Knowledge Distillation. Peng et al. arXiv 2019

**Questions:**

Comments:

1. Typo: may prove may provide (page 3 top paragraph).

2. The authors have cited some works from the field of psychology as the source of their motivation. While it is fine to refer to works from other fields, one should cite them only when there is something substantive to talk about. Right now, the section talking about those works (Section 1, paragraph 3) seems to have not much context/relevance to the particular problem that the authors are solving.